# Mesenchymal Stem Cells Pretreated with Collagen Promote Skin Wound-Healing

**DOI:** 10.3390/ijms24108688

**Published:** 2023-05-12

**Authors:** Zheng Kou, Balun Li, Aili Aierken, Ning Tan, Chenchen Li, Miao Han, Yuanxiang Jing, Na Li, Shiqiang Zhang, Sha Peng, Xianjun Zhao, Jinlian Hua

**Affiliations:** College of Veterinary Medicine, Shaanxi Centre of Stem Cells Engineering & Technology, Northwest A & F University, Yangling, Xianyang 712100, China; 2020055520@nwafu.edu.cn (Z.K.); libalun@nwafu.edu.cn (B.L.);

**Keywords:** mesenchymal stem cells, collagen, skin wound, repair

## Abstract

The existing treatment modalities for skin injuries mainly include dressings, negative-pressure wound treatment, autologous skin grafting, and high-pressure wound treatment. All of these therapies have limitations such as high time cost, the inability to remove inactivated tissue in a timely manner, surgical debridement, and oxygen toxicity. Mesenchymal stem cells have a unique self-renewal ability and wide differentiation potential, and they are one of the most promising stem cell types in cell therapy and have great application prospects in the field of regenerative medicine. Collagen exerts structural roles by promoting the molecular structure, shape, and mechanical properties of cells, and adding it to cell cultures can also promote cell proliferation and shorten the cell doubling time. The effects of collagen on MSCs were examined using Giemsa staining, EdU staining, and growth curves. Mice were subjected to allogeneic experiments and autologous experiments to reduce individual differences; all animals were separated into four groups. Neonatal skin sections were detected by HE staining, Masson staining, immunohistochemical staining, and immunofluorescence staining. We found that the MSCs pretreated with collagen accelerated the healing of skin wounds in mice and canines by promoting epidermal layer repair, collagen deposition, hair follicle angiogenesis, and an inflammatory response. Collagen promotes the secretion of the chemokines and growth factors associated with skin healing by MSCs, which positively influences skin healing. This study supports the treatment of skin injuries with MSCs cultured in medium with collagen added.

## 1. Introduction

The skin is the largest organ in the body and has many different functions. In addition to acting as a physical barrier, it also has immune and sensory properties [1]. It is the first line of defense against infection and damage. In addition, it plays an important role in the regulation of body temperature and vitamins [2]. The skin is a key organ that protects the body’s internal tissues. It is extremely vulnerable when damaged by trauma, disease, etc. Especially when skin damage occurs in patients with genetic diseases such as diabetes and chain cell disease, infected wounds are extremely difficult to heal. At present, the treatments for skin damage mainly include dressings, negative-pressure wound therapy, autologous skin grafting, and high-pressure wound therapy [3]. These therapies have limitations such as high time cost, inability to remove devitalized tissue in a timely manner, surgical debridement, and oxygen toxicity. There is, therefore, a need for more effective treatments for repairing skin damage. In order to circumvent the limitations of traditional methods, the application of stem cells has become a promising approach for both acute and chronic wounds [4].

MSCs belong to stromal cells and have a strong ability to self-renew and activate multi-lineage differentiation [5]. MSCs have the potential to differentiate into bone cells, chondrocytes, fat cells, muscle cells, and other cells [6]. MSCs proliferate in vitro as adherent cells with a fibroblast-like morphology [7]. MSCs can spontaneously migrate to an injured area and differentiate into the desired tissues spontaneously [8,9]. Currently, mesenchymal stem cells are considered to be one of the most promising stem cell types for cell therapy. MSCs have a unique self-renewal ability and broad differentiation potential, which provides them with great application prospects in the field of regenerative medicine [10]. However, there are still some problems with current MSC transplantation therapy, including the uneven quality of MSCs used for treatment, the incomplete quality evaluation system of MSCs, and the unclear treatment mechanism [11]. The source and quality of cells have become the main factors limiting the use of MSCs in clinical therapy. There is an urgent need to further explore the mechanism of MSC therapy to provide strong evidence for the application of MSCs in clinical medicine [12].

Collagen is the most abundant protein in mammals, and most types of collagen form supramolecular assemblies and are deposited in the extracellular matrix [13]. Collagen acts by promoting the molecular structure, shape, and mechanical properties of tissues, contributing to their mechanical properties, organization, and shape. Collagen interacts with cells through several receptor families and regulates their proliferation, migration, and differentiation [14]. Collagen is the primary extracellular matrix (ECM) molecule that self-assembles into intersecting striated fibrils, providing support for cell growth, and it is responsible for the mechanical elasticity of connective tissue [15].

Therefore, it is worth exploring how collagen can better combine with MSCs to achieve the purpose of promoting wound repair, especially for diabetes patients with skin damage. This study investigated the effects of MSCs combined with collaged to treat diabetic mice and canine skin damage models.

## 2. Results

### 2.1. Collagen Enhances MSC Activity

A proteomic analysis of the culture medium of normal cultured mesenchymal stem cells found that the content of collagen accounted for a large proportion of the supernatant (Figure 1A). By adding different doses of collagen, it was determined that 10 μg/mL of collagen has a significant promoting effect on MSCs (Figure 1C,D). As shown by EdU staining and cell growth curve results, the proliferation rate of cells in the collagen group was faster than that of the normal ADMSC group (Figure 1B,E,F). The results showed that the addition of collagen to a common medium could promote the proliferation and growth of MSCs and improve the activity of MSCs.

### 2.2. Collagen Enhanced the Repair Effect of the MSCs on Skin Damage in Mice

Circular skin excisions 1.1 cm in diameter on the backs of mice were treated. The allogeneic and autologous skin healing of the mice were recorded (Figure 2A,C). The wound areas in all groups decreased with time. In the allogeneic and autologous experiments, the PBS group had the slowest healing rate, and the Col + ADSC group healed the fastest, and wound repair was completed in 11 days and 13 days, respectively. The results showed that the MSCs with added collagen could shorten the wound-healing time and promote the repair of skin damage in mice.

### 2.3. Collagen Promoted the MSCs to Repair Skin Damage in Multiple Ways

The HE staining results showed more newly formed blood vessels and hair follicles in the Col + ADSC group and the ADSC group than in the PBS group, and the recovery effect of the epidermis in the Col + ADSC group was better (Figure 3A,B). In the Masson staining, the amount of collagen fibers in the blue section of the PBS group was very small. The color analysis of the Col + ADSC group, ADSC group, and PBS collagen deposition was compared, and the collagen deposition of the Col + ADSC and ADSC groups compared to PBS group was analyzed, and the collagen deposition of the Col + ADSC group was higher than that of the ADSC group. The collagen fiber repair was better in the Col + ADSC group (Figure 3C,D).

In the immunohistochemistry, there were fewer PCNA-positive cells in the PBS group, and the largest number of brown cells was found in the Col + ADSC group, indicating that in the new skin, the cell proliferation effect was the best (Figure 3E). In the CD68 immunofluorescence staining of the skin sections, where CD68 primarily labeled the M1 macrophages and where the M1 macrophages mainly functioned as pro-inflammatory, killing microorganisms with a role in the early stages of trauma, the number of M1 macrophages in the Col + ADSC and ADSC groups decreased, indicating that the new skin state had changed from pro-inflammatory to anti-inflammatory, and the wound-healing stage was faster. Among the staining samples, the Col + ADSC group had the lowest number of M1 macrophages and the most obvious anti-inflammatory effect (Figure 3F). The results showed that collagen promoted the MSCs to repair the skin damage in terms of epidermal layer repair, collagen deposition, cell proliferation, and the inflammatory response.

### 2.4. Collagen Enhanced the Repair Effect of the MSCs on Normal Dog Skin Damage

Circular skin excisions of 3 cm in diameter were performed on the dogs’ abdomens, and the skin-healing rates were recorded (Figure 4A). The wound areas of all groups decreased with time. The PBS group had the slowest healing rate; the Col + ADSC group healed the fastest; and the wounds of the Col + ADSC group had basically healed at 11 days (Figure 4B). HE staining was performed on samples from the skin that had healed after 11 days. There were more newly formed blood vessels and hair follicles in the Col + ADSC and ADSC groups than in the PBS group, and the thicknesses of the epidermis layers were the lowest in the Col + ADSC group, though the recovery effect was better (Figure 4C,E). In the Masson staining of the skin that had healed after 11 days, it was found that the amount of collagen fibers in the blue section of the PBS group was very small. The color analyses of the Col + ADSC and ADSC groups and the PBS collagen deposition were compared, and the collagen depositions of the Col + ADSC and ADSC groups compared to that of the PBS group were analyzed. The collagen deposition in the Col + ADSC group was higher than that in the ADSC group, and the collagen fibers were better repaired (Figure 4D,F). The results showed that after adding remote collagen to the medium, the MSCs could shorten the wound-healing time in a normal canine, promote the formation of new skin blood vessels and hair follicles, and promote epidermal layer repair and collagen fiber production, thereby promoting the repair of the skin damage.

### 2.5. Collagen Can Enhance the Repair Effect of MSCs on Skin Damage in Diabetic Canines

Four circular skin excisions of the same areas were performed on the abdomens of diabetic canines, and the skin-healing rates were recorded (Figure 5A). The wound areas of all groups decreased with time. The PBS group had the slowest healing rate; the Col + ADSC group healed the fastest; and the wounds of the Col + ADSC group had basically healed at 13 days (Figure 5B). HE staining was performed on samples from the skin that had healed after 13 days, and there were more newly formed blood vessels and hair follicles in the Col + ADSC and ADSC groups and fewer in the PBS group, and the thicknesses of the epidermis layers were the lowest in the Col + ADSC group, though the recovery effect was better (Figure 5C,D). Masson staining was performed on samples from the skin that had healed after 13 days, and the amount of collagen fibers in the blue section of the PBS group was very small. The color analyses of the Col + ADSC and ADSC groups and the PBS collagen deposition were analyzed, and the collagen depositions of the Col + ADSC and ADSC groups were compared with that of the PBS group. The collagen deposition in the Col + ADSC group was higher than that in the ADSC group, and the collagen fibers were better repaired (Figure 5E,F). The results showed that after adding remote collagen to the medium, the MSCs could shorten the wound-healing time in the diabetic canines, promote the formation of new skin blood vessels and hair follicles, and promote epidermal layer repair and collagen fiber production, thereby promoting the repair of the skin damage.

### 2.6. Canine ADSC Differential Expression Analysis Affected by Collagen

KEGG pathway analysis showed that differentially expressed genes were significantly enriched in 40 pathways, and there were many pathways related to coagulation and cell adhesion in the KEFG pathway, such as complement and coagulation cascades. It is also enriched with signaling pathways related to cell cycle and regulation of inflammation, such as the cell cycle (Figure 6A,B).

## 3. Discussion

To date, many studies have investigated the role of MSCs in four overlapping stages of skin wound-healing: hemostasis (coagulation), inflammation (single-cell infiltration), proliferation (epithelialization, fibrogenesis, and angiogenesis), and maturation (collagen deposition and scarring), as well as their the roles of tissue formation [16]. MSCs can participate in all stages of skin wound-healing, promoting wound-healing and reducing scarring during skin treatments. MSCs migrate to the site of skin damage, inhibit inflammation, and increase fibroblasts, epidermal cells, and the proliferation and differentiation potential of endothelial cells [17,18,19]. However, there are still many problems with the use of MSCs for treatment, and various unfavorable factors in vitro and outside affect the cell status of MSCs, reduce the activity of MSCs, and hinder their therapeutic effects [20,21]. Long-term in vitro cultures have resulted in decreased MSC proliferation capacity, senescence, and morphological changes [20,22]. In addition, 80–90% of ADMSCs died within 72 h after transplantation [22,23,24]. The low survival and proliferation rate of MSCs after transplantation is mainly due to the lack of required nutrients or growth factors for MSCs in vivo. In addition, adverse factors, such as oxidative stress and chronic inflammation in vivo, affect the survival of MSCs in vivo [25]. To date, studies have tried to promote the viability of MSCs in vivo and in vitro, but the effect is still unsatisfactory, and studies aimed at promoting cell viability are lacking [23,25,26,27,28,29].

Collagen can play a structural role by promoting the molecular structure, shape, and mechanical properties of tissues [14]. The addition of collagen to cell cultures has many advantages as collagen can promote cell proliferation and reduce the cell-doubling time [30]. Further, a collagen matrix can effectively protect MSCs from the oxidative stress-induced cell death that may occur in vivo during ischemia [31]. In this study, we added collagen to an MSC culture medium. The results showed that the collagen promoted the viability of the MSCs cultured in vitro. Transcriptome sequencing data analysis confirmed that collagen can not only promote the ADSC cell cycle, lipid synthesis, and cell proliferation and division but also confirmed that it can also activate the complement and coagulation cascade pathway. After the activation of the coagulation cascade pathway, other enzyme cells in the blood are activated in turn, resulting in the original signal being amplified, promoting the activation of coagulation factors through the endogenous coagulation cascade reaction and preventing blood vessels from bleeding or blood loss at the moment of rupture, which is an important response of the body’s coagulation mechanism, thus promoting the healing of the skin.

We then used the MSCs cultured with collagen to treat skin damage in mice and canines. We found that the MSCs significantly shortened the healing period after damage, and the repaired epidermis, collagen deposition, hair follicle angiogenesis, and inflammatory response had significant promoting effects. Especially in the treatment of diabetic canines, where the wounds appeared to be slightly infected, the collagen-cultured MSCs still showed good therapeutic efficacy.

## 4. Materials and Methods

### 4.1. Animals and Establishment Skin Wound

Male Kunming mice (26 ± 5 g) aged six weeks were obtained from the experimental animal center of the DOSSY EXPERIMENTAL ANIMALS CO., LTD. (Xi’an, China). Mice were allowed free access to distilled water and standard food and were housed in a room under conditions of maintained temperature (24–26 °C) and humidity (69–71%) as well as a 12:12 h light–dark cycle.

Six Chinese pastoral dogs, weighing 5–6.5 kg, male, were purchased from the dog and cat trading market, with a physical examination before the start of the experiment; all indicators were carried out normally; all experimental dogs were kept in a ventilated and clean environment, given enough food and water. Three dogs were fed for 4 weeks and then intravenously injected with STZ 25 mg/kg/day for a total of 3 days [12], and the dogs were made to fast for 24 h before injection. One week after modeling, blood glucose levels were measured to determine diabetes modeling.

All experimental protocols were carried out according to the guidelines established by the Chinese National Standard GB/T35892-2018 (guidelines for the ethical review of laboratory animal welfare), submitted, and previously approved by the Ethics Committee on the Use of Animals of the Northwest A&F University, approval number (NWAFU.No20234580d0600601[200]).

### 4.2. Establishment of Mice Skin Wound

The dorsal 12 male mice were subjected to 1.1 cm circular skin resection and were divided into the following three groups: (1) PBS, (2) ADSC, and (3) Col + ADSC. Three rounds of 1.1 cm circular skin excision were performed on the dorsal of 6 male mice for mouse piedu injury allogeneic experiments. The wounds were divided into the following 3 groups: (1) PBS, (2) ADSC, and (3) Col + ADSC, for self-experiments on skin injury in mice. After the model was established, cells were injected subcutaneously around the wound.

Mouse skin injury allogeneic experiments and normal dog skin injury experiments were collected for 11 days of healing skin. Samples from the mouse skin injury autologous experiments and the diabetic dog skin injury experiments were collected for 13 days to heal the skin. The above skin is used for tissue sections.

### 4.3. Collagen Preparation

Type-1 collagen taken from the Achilles tendons of cattle was purchased from China Shanghai Yuanye Biotechnology Co., Ltd Next (Shanghai, China), 5.72 mL of glacial acetic acid was dissolved in 994.28 mL of distilled water and was then shaken and mixed well. Then, 0.1 mg of collagen powder was weighed out, and 0.1 M acetic acid was added to dissolve it. In the following step, 1% (*w*/*v*) collagen was added to a magnetic stirrer and was stirred at room temperature for 1–3 h until the collagen was completely dissolved. On the bottom of the inside of a sterile glass bottle, a layer of chloroform was spread, and the collagen solution was transferred to the upper chloroform layer. The volume of the chloroform was 10% that of the collagen solution, and shaking and stirring were avoided. The solution was left to rest overnight at 2–8 °C. The upper collagen solution was filtered through a 0.45 filter and transferred to a sterile container (aseptic operation) to obtain a collagen stock solution. The collagen stock solution was added to the culture medium to obtain collagen culture medium (Col culture medium) to culture ADSC and obtain Col-ADSC.

### 4.4. Establishment of Canine Skin Wound

Three 3 cm-diameter circular skin excisions were performed on the abdomen of three 1-year-old healthy male canines. The wounds were divided into the following three groups: (1) PBS, (2) ADSC, and (3) Col + ADSC. Four 1.5 cm circular skin excisions were performed on the abdomen of three 1-year-old diabetic male dogs, and the wounds were divided into the following four groups: (1) PBS, (2) ADSC, (3) Col, and (4) Col + ADSC.

### 4.5. Adipose Tissue Collection and ADSC Isolation

The canines were given general zoletil (Virbac group, Carros, France) injection anaesthesia, and then, adipose tissue was aseptically harvested from abdominal subcutaneous fat. Canine adipose tissue was minced using a sterile scalpel and surgical scissors, placed in a 50-mL tube with an equal volume of pre-heated PBS, and agitated for 30 s to wash. The tissues were allowed to separate into phases for 3 min, and then the infranatant solution was removed. The tissue was rinsed with PBS to remove the erythrocytes and white blood cells. The tissue was shaken for 45 s and then left to float to the top. The sample was rinsed until the infranatant was clear. Collagenase type I solution (Roche Diagnostics, Mannheim, Germany) was added per volume of adipose tissue, and then, it was placed in a 37 °C water bath and vortexed every 15 min. It was then vortexed for 15 s to thoroughly mix cells and then centrifuged at 252× *g* for 5 min. The supernatant was then removed. Two to three microliters of liquid above the cell pellet were left behind so that the stromal vascular fraction was not disturbed. The cells were resuspended in 1% BSA solution, centrifuged at 252× *g* for 5 min, and then removed from the supernatant. The pellet was then resuspended in a known volume of cell culture medium and centrifuged at 1500 rpm for 5 min. The supernatant was then discarded, and the pellet was resuspended in an equal volume of red cell lysis buffer and incubated for 5 min. The cells were counted using a hemocytometer. The cells were plated at an appropriate density in complete cell medium [32].

### 4.6. Cell Isolation and Culture

MSCs are derived from abdominal adipose tissue from two 5-month-old male hybrid canines. Detailed MSC isolation steps and MSC identification were described in our previous report [32]. Cells were grown in α-MEM (Invitrogen, Carlsbad, CA, USA) complete medium containing 10% fetal bovine serum (Gibco, New York, NY, USA) at 37 °C in a 5% CO incubator [33,34]. Passage 1:3 when cells are attached to the bottom of the plate at approximately 80%. The fourth-generation cells were treated and used for cell therapy. Collagen was added to the medium 72 h before transplantation, sample collection, and staining.

### 4.7. Optimization of the Concentration of MSCs in Collagen Culture

MSCs were seeded into 48-well plates. Collagen solution (0.1 μg/μL) was added to the culture medium at doses of 12.8, 19.2, 25.6, 32, 38.4, 46.2, and 51.2 μL, according to the content of 4, 6, 8, 10, 12, 14, and 16 μg/mL.

### 4.8. Histological Analysis

For each group of mice, the liver, pancreas, and white adipose tissue were fixed overnight in 4% neutral formalin solution before being embedded in paraffin. Tissues were cut into 5 μm-thick sections and stained with hematoxylin and eosin (HE) Masson staining [35].

### 4.9. Cell Growth Curve

ADSCs and Col-ADSCs cells were digested, centrifuged, resuspended, counted, and seeded into 12-well plates at 5 × 10^3^ cells per well, and α-MEM (+) was used to change the medium every day, every 24 h to digest and count the cells in 3 wells in each of the two groups, and to change the medium in the other wells until the 8th day. After 8 days, the obtained cell counting results are drawn with the time as the abscissa and the number of cells as the ordinate to draw the cell growth curve [12].

### 4.10. Immunofluorescent, Immunohistochemical, and EdU Staining

The expression levels of PCNA (1:200, rabbit IgG, Bioss, Beijing, China), CD68 (1:200, rabbit IgG, Bioss, China), and PCNA (1:100, Immunoway Biotechnology, Plano, TX, USA), were detected by immunofluorescent, immunohistochemical, and EdU analysis as previously described [36,37].

### 4.11. Transcriptome Sequencing Analysis

Canine ADSC was divided into the NC and Col groups, and the normal α-MEM culture medium and the α-MEM culture medium with collagen were cultured to the third generation. Cells were collected for transcriptomic sequencing analysis [38,39].

The obtained RNA was delivered to Lianchuan Biotech for library construction and subsequent RNA sequencing (RNA qualification standards: OD260/280 = 1.18~2.2, OD260/230 ≥ 2.0, RIN ≥ 6.5, 28S:18S ≥ 1.0), and the original data and gene expression matrix were returned for subsequent analysis. The Lianchuan cloud tool platform was used to enrich each group of differential genes using GO and KEGG.

### 4.12. Statistical Analysis

Statistical analyses were performed with SPSS version 19.0 (IBM Corporation, Chicago, IL, USA) software. All the experimental data were shown as mean ± SD, and the results were examined with one-way analysis of variance (ANOVA). A *p* value < 0.05 indicated the statistical significance observed in the comparison.

## 5. Conclusions

In this study, collagen was used as an additive for the in vitro culture of MSCs. Collagen can promote the adhesion and proliferation of MSCs. Collagen can upregulate the gene pathway of ADSC in skin injury repair in dogs, especially the coagulation cascade, and promote blood clotting after injury. This further promotes wound healing. The MSCs cultured with collagen were used to treat mouse and canine skin damage, as they can play an important role in epidermal repair, collagen deposition, hair follicle angiogenesis, inflammatory response, etc., in the process of skin healing. Moreover, they also had a good effect on the treatment of skin damage in diabetic canines. Collagen can promote MSCs and skin repair. Healing-related chemokines and growth factors are secreted by collagen, and it also has a good promoting MSC effect on damaged skin and other tissues through immune homeostasis [30,40]. However, the preliminary preparation of mesenchymal cells as a treatment method for skin injury is still cumbersome, and the preparation of fast mesenchymal stem cell preparations is an important task to optimize the treatment process in the next step.

The healing of skin wounds in mice and dogs is generally consistent with the healing process of human skin, all involving inflammatory response, neoangiogenesis, collagen deposition, and scar tissue production. The effect of collagen to promote ADSC in the repair of skin injuries in mice and dogs has been verified and still has reference significance for human skin tissue repair; especially for diabetic dogs, the treatment effect is of great significance for diabetic patients such as skin ulceration and wound difficulty healing.

## Figures and Tables

**Figure 1 ijms-24-08688-f001:**
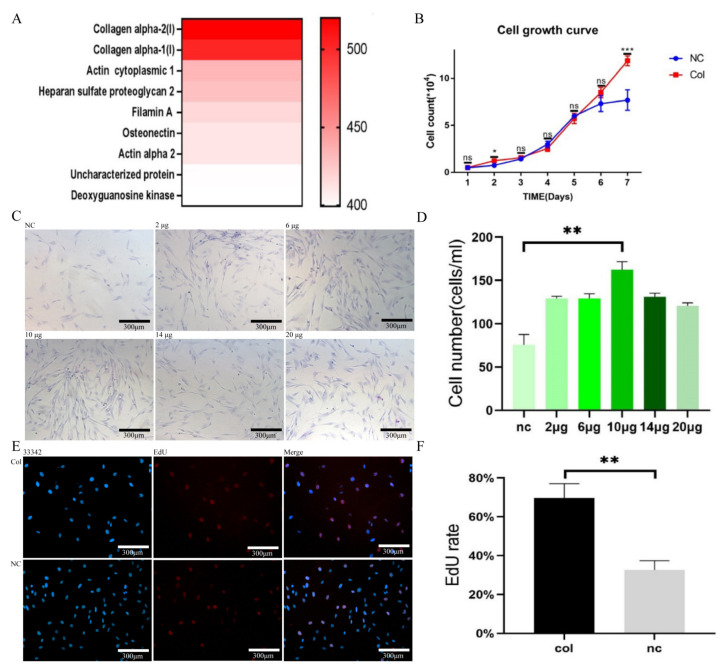
Collagen can enhance the vitality of MSCs. (**A**) Proteomic analysis of MSC supernatants. (**B**) Cell growth curve. (**C**) Giemsa staining. (**D**) Quantitative analysis of the Giemsa staining. (**E**) EdU staining. (**F**) EdU staining quantification. The data shown are means ± SDs; *n* = 3 per group; ns > 0.05, * *p* < 0.05, ** *p* < 0.01, and *** *p*< 0.001, as determined by a repeated-measures ANOVA test.

**Figure 2 ijms-24-08688-f002:**
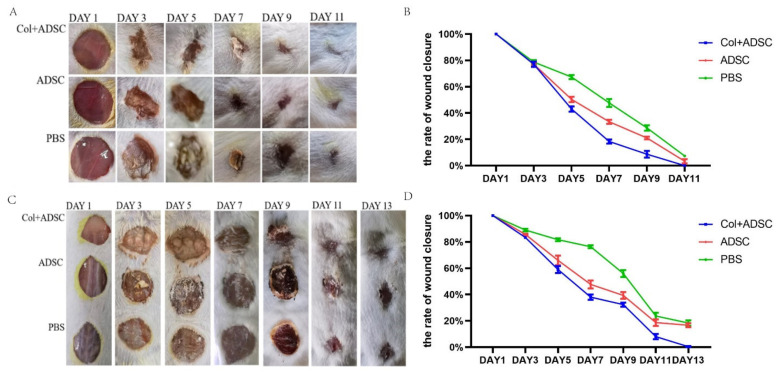
MSCs cultured with collagen can promote the repair of skin damage in mice. (**A**) Wound-healing records for the mouse skin injury allogeneic experiments. (**B**) Quantitative analysis of the wound-healing rates from the mouse skin injury allogeneic experiments. (**C**) Mouse skin injury autologous experimental wound-healing records. (**D**) Quantitative analysis of the wound-healing rates from the mouse skin injury autologous experiments. The data shown are means ± SDs, and *n* = 3 per group.

**Figure 3 ijms-24-08688-f003:**
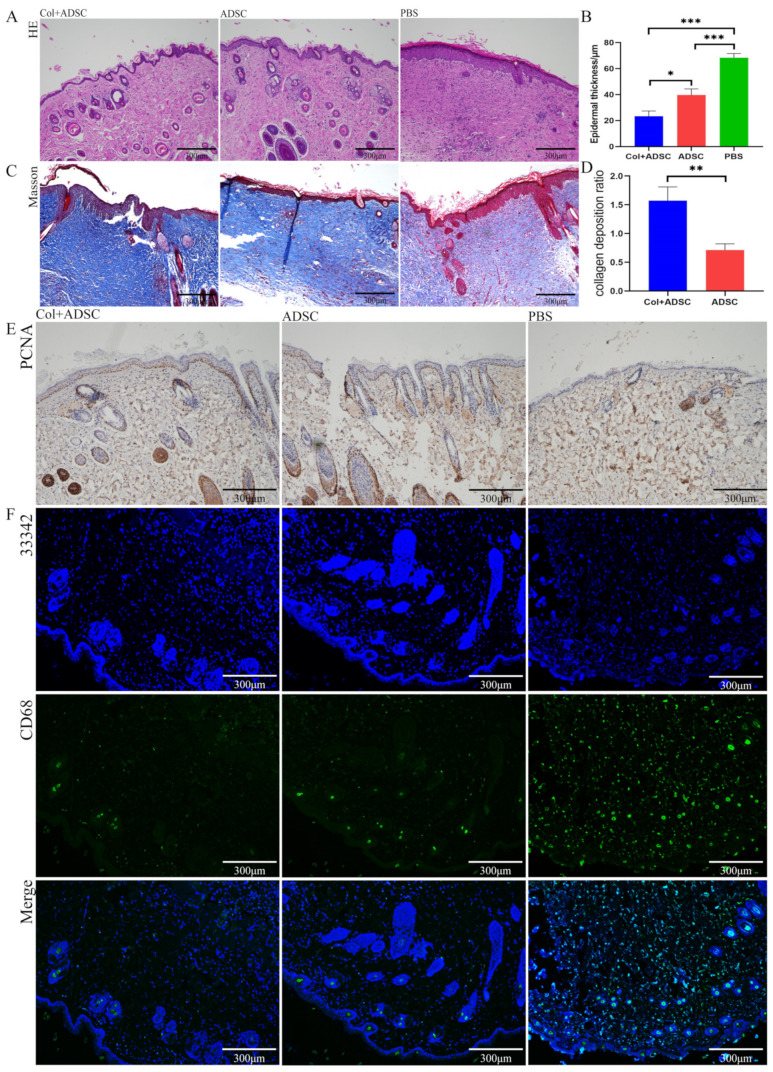
Collagen promotes MSCs to repair skin damage in multiple ways. (**A**) HE staining of 11-day-old mouse healing skin. (**B**) Quantitative analysis of epidermal thickness by HE staining. (**C**) Masson staining of 11-day-old mouse healing skin. (**D**) Quantitative analysis of collagen deposition by Masson staining. (**E**) Immunohistochemistry of PCNA for 11-day-old mouse healing skin. (**F**) Immunofluorescence staining of 11-day-old mouse healing skin. The data shown are means ± SDs; *n* = 3 per group; * *p* < 0.05, ** *p* < 0.01, and *** *p* < 0.001, as determined by a repeated-measures ANOVA test.

**Figure 4 ijms-24-08688-f004:**
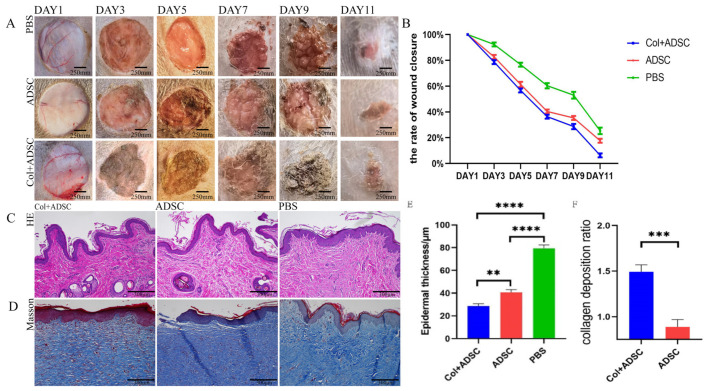
MSCs cultured with collagen can promote healing in normal canine skin damage. The addition of collagen-cultured MSCs can promote the repair of normal canine skin damage. (**A**) Records of wound-healing in the normal dog skin injury experiments. (**B**) Quantitative analysis of the skin injury wound-healing rates. (**C**) HE staining of 11-day-old canine healing skin. (**D**) Masson staining of 11-day-old canine healing skin. (**E**) Quantitative analysis of the epidermal thickness using HE staining. (**F**) Quantitative analysis of collagen deposition using Masson staining. The data shown are means ± SDs; *n* = 3 per group; ** *p* < 0.01, *** *p* < 0.001, and **** *p* < 0.0001, as determined by a repeated-measures ANOVA test.

**Figure 5 ijms-24-08688-f005:**
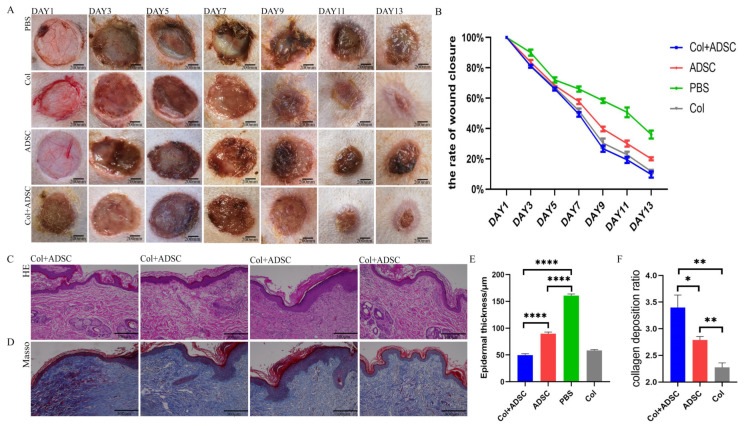
MSCs cultured with collagen can promote the repair of skin damage in diabetic canines. (**A**) Experimental wound-healing records of the skin injuries in diabetic canines. (**B**) Quantitative analysis of the skin injury wound-healing rate. (**C**) HE staining of 13-day-old canine healing skin. (**D**) Masson staining of 13-day-old canine healing skin. (**E**) Quantitative analysis of the epidermal thicknesses by HE staining. (**F**) Quantitative analysis of collagen deposition by Masson staining. The data shown are means ± SDs; *n* = 3 per group; * *p* < 0.05, ** *p* < 0.01, and **** *p* < 0.0001, as determined by a repeated-measures ANOVA test.

**Figure 6 ijms-24-08688-f006:**
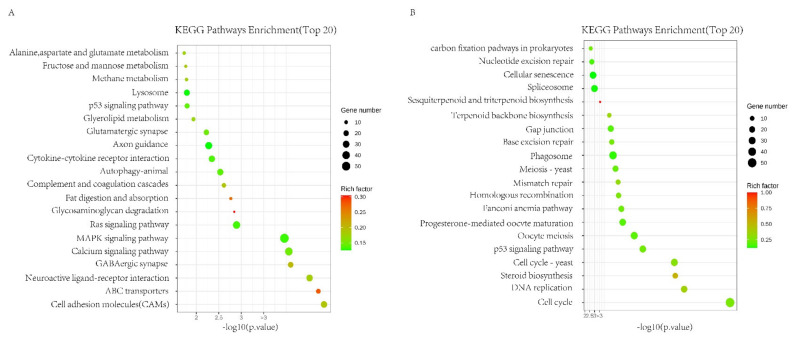
Canine ADSC differential genetic analysis affected by collagen. (**A**) ADSC & Col + ADSC differential expression gene KEGG enrichment analytical upregulation pathway. (**B**) ADSC & Col + ADSC differential expression gene KEGG enrichment analytical downregulation pathway.

## Data Availability

All data generated and analyzed during this study are included in this published article.

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
