# Peer review of "Mesenchymal Stem Cells Pretreated with Collagen Promote Skin Wound-Healing"

_ijms, 2023, doi:10.3390/ijms24108688_

Round 1

Reviewer 1 Report

“Mesenchymal stem cells pretreated with collagen promote skin wound-healing”. The authors found that the MSCs pretreated with collagen accelerated the healing of skin wounds in mice and canines by promoting epidermal layer repair, collagen deposition, hair follicle angiogenesis, and the inflammatory response. The study is interesting and well designed. However, the following points that the authors should address  before acceptance.

1. Citation formats need to be carefully revise as IJMS guidelines.

2. Line 28-29, I suggest to remove “In addition, a similar effect was observed for the treatment of skin lesions in diabetic canines.”

3. The authors should add more references in the Materials and methods

4. The species of canines should be listed.

5. The collagen usage methods need to be detailed.

The English is readable, extensive editing may improve the quality of manuscript.

Reviewer 2 Report

Dear authors.

I read an interesting paper today on the use of matrix cells in wound healing. The research methods used as well as the study design are appropriate. The use of a rodent and a dog model in the study seems to be a good concept, although in my country it is said that "a wound heals like on a dog". Therefore, it is worth considering yet another experimental model or performing an experimental therapy in dogs with difficult-to-heal wounds.

After reading, however, I have the following comments:

1. however, I would like to ask for a brief addition in the methodology of the isolation conditions and the method of cell characterization, including the percentage of stem cells obtained.

2. What was the final concentration of collagen in the culture medium, which exact calegene was used, origin, manufacturer. Was there bovine serum in the culture medium, at what concentration and of what quality (biotechnological, pharmaceutical, research).

3. Conclusions too short. Why the collagen supplement works. A diagram should be added to the paper to show the proposed mechanism of action of collagen with cells.

Round 2

Reviewer 2 Report

Dear authors,

would like to thank you sincerely for your corrections to the paper and the questions you have answered. I wish you success in your further research.